

# Reconstructing the transmission dynamics of varicella in Japan: an elevation of age at infection

Ayako Suzuki and Hiroshi Nishiura

School of Public Health, Kyoto University, Kyoto, Kyoto, Japan

## ABSTRACT

**Background:** In Japan, routine two-dose immunization against varicella has been conducted among children at ages of 12 and 36 months since 2014, and the vaccination coverage has reached around 90%. To understand the impact of routine varicella vaccination, we reconstructed the epidemiological dynamics of varicella in Japan.

**Methods:** Epidemiological and demographic datasets over the past three decades were analyzed to reconstruct the number of susceptible individuals by age and year. To estimate the annual risk of varicella infection, we fitted a balance equation model to the annual number of cases from 1990 to 2019. Using parameter estimates, we reconstructed varicella dynamics starting from 1990 and modeled future dynamics until 2033.

**Results:** Overall varicella incidence declined over time and the annual risk of infection among children younger than 10 years old decreased monotonically starting in 2014. Conversely, varicella incidence among teenagers (age 10 to 14 years) has increased each year since 2014. A substantial number of unvaccinated individuals born before the routine immunization era remained susceptible and aged without contracting varicella, while the annual risk of infection among teenagers aged 10 to 14 years increased starting in 2011 despite gradual expansion of varicella vaccine coverage. The number of susceptible individuals decreased over time in all age groups. Modeling indicated that susceptibility rates among pre-school children aged 1 to 4 years will remain low.

**Conclusion:** Routine varicella vaccination has successfully reduced infections in pre-school and early primary school age children, but has also resulted in increased infection rates among adolescents. This temporary increase was caused both by the increased age of susceptible individuals and increased transmission risk among adolescents resulting from the dynamic nature of varicella transmission. Monitoring susceptibility among adolescents will be important to prevent outbreaks over the next decade.

Corresponding author
Hiroshi Nishiura,
nishiurah@gmail.com

## INTRODUCTION

Varicella-zoster virus (VZV) is a human alphaherpesvirus with a double-stranded DNA genome. VZV is the causative agent of varicella (chickenpox) and herpes zoster. Primary infection with VZV results in varicella, which is characterized by an itchy vesicular rash. Varicella causes mild to moderate illness in healthy children but symptoms can be serious, especially in infants, adolescents, adults, and immunocompromised individuals. VZV enters a latent state in dorsal root ganglia and subsequent reactivation causes herpes zoster. Older and immunosuppressed individuals are at increased risk of herpes zoster (*Arvin, 1996*; *Gershon et al., 2015*).

A live attenuated varicella vaccine, initially derived from the Oka strain, was developed in Japan in 1994 (*Takahashi et al., 1974*). Prior to the introduction of varicella vaccines, varicella was a universal childhood disease worldwide. A varicella vaccination program was first introduced in the United States in 1995 and varicella incidence declined by 90% over the next decade (*Guris et al., 2008*). However, single dose vaccination was considered insufficient to prevent VZV transmission in children, and high numbers of breakthrough infections led to adoption of routine two-dose vaccination starting in 2007 (*Lopez et al., 2006*; *Gao et al., 2010*). Based on data from the *World Health Organization's (2018)* Vaccine Preventable Diseases Monitoring System, a total of 51 countries and regions had implemented one- or two-dose universal varicella vaccination as of 2021. A meta-analysis of published studies from 1995 to 2014 showed that one-dose and two-dose varicella vaccination was 81% and 92% effective in preventing varicella, respectively (*Marin et al., 2016*).

Although the varicella vaccine was commercially available in Japan starting in 1987, it was not rapidly integrated into routine vaccination programs and the vaccination rate remained around 40% during the 1990s (*Yoshikawa, Kawamura & Ohashi, 2016*). Subsequently, vaccine coverage gradually increased starting around 2010 as some local governments initiated subsidy program for varicella vaccination. The Japan Pediatric Society recommended two-dose varicella immunization in 2012. In 2014, varicella vaccination coverage (received at least one-dose of varicella vaccine) reached around 90% after a two-dose varicella immunization program for children aged 12 to 36 months was introduced, and by 2019 vaccination had resulted in an 80% reduction in varicella incidence (*National Institute of Infectious Diseases, 2018*).

Theoretical concerns regarding the introduction of universal varicella vaccination included increased age at infection following mass vaccination and potential for increased incidence of herpes zoster (*Gidding et al., 2005*; *Karhunen et al., 2010*). Varicella vaccine is not included as a routine childhood immunization in the United Kingdom as of 2021 (*National Health Service, 2019*). Varicella in adults can be serious and even result in death in some individuals. The overall case-fatality risk (CFR) of varicella from 1990 to 1994 was 2.6 per 100,000 infections with the highest CFR observed in those older than 20 years. The CFR among adults aged more than 20 years was 21.3, 25 times higher than that among children aged 12 months to 4 years (*Meyer et al., 2000*). Various modeling studies have been carried out to evaluate the population impact of universal varicella vaccination.

*Halloran et al. (1994)* showed that the number of older susceptible children increased after initiation of a varicella vaccination program, and that this effect became more apparent as vaccine coverage increased. Many followers assessed the impact of universal varicella immunization on the incidence of varicella (*Brisson et al., 2000*; *Karsai et al., 2020*; *Pawaskar et al., 2021*; *Suh et al., 2021*; *van Hoek et al., 2011*).

Despite numerous modelling efforts, it remains unclear how the numbers of susceptible individuals of a given age will impact the long-term epidemiological dynamics of varicella. Almost seven years have elapsed since the initiation of routine varicella vaccination in Japan. Monitoring changes in the fraction of susceptible individuals by birth cohort could help to project future varicella dynamics. The aim of this study was to assess temporal changes in the Japanese population susceptible to varicella. Using reported number of cases from 1990 to 2019, we estimated the risk of infection as a function of time and age and reconstructed epidemiological dynamics until 2033.

## MATERIALS AND METHODS

### Epidemiological data

Three different types of data were used: (i) age-specific annual incidence of varicella (*i.e.*, annual number of notified chickenpox cases by age group), (ii) vaccination coverage, and (iii) number of newborns. Varicella incidence was based on notifications to the National Epidemiology Surveillance for Infectious Diseases (NESID) system in compliance with the Communicable Disease Prevention Law until March 1999 and subsequently the Infectious Diseases Control Law. Varicella case was defined as presentation to medical services with the following symptoms: (1) sudden generalized development of serous papules and vesicles and (2) co-existence of rashes in different stages (papules, vesicles, and crusts) (*Ministry of Health, Labour & Welfare of Japan, 2021a*). Varicella is classified as a category V disease, a subset of which including varicella are monitored by sentinel surveillance systems. Varicella is a pediatric disease in Japan, and physicians at approximately 3,000 pediatric sentinel sites (representing approximately 10% of pediatric clinics and hospitals in Japan) notified cases to NESID on a weekly basis. Reporting rate in the NESID system (*i.e.*, the proportion of notified cases included in surveillance out of all infections) was calculated as the cumulative number of cases divided by the number of newborns in each cohort. Since the surveillance system was revised along with the change in law in April 1999, reporting rate was separately calculated before and after 1999 (see Fig. S1).

Routine assessment of varicella vaccination coverage was not conducted before the introduction of the routine varicella vaccination program in 2014. Instead, we used vaccination coverage estimated from annual vaccine sales divided by numbers of births in the previous year (*Ozaki, 2013*). Vaccination coverage after 2014 was extracted from the database of the National Epidemiological Surveillance of Vaccine-Preventable Diseases (NESVPD), which regularly investigates the seroprevalence and vaccination rates for infectious diseases subject to routine vaccination (*Ministry of Health, Labour & Welfare of Japan, 2021b*).

To model the future dynamics of varicella in Japan, we used the number of newborns after 2021 from the Population Projections for Japan (2017) from 2016 to 2065 (*National Institute of Population & Social Security Research, 2017*).

## Mathematical model

In the following model, we consider the depletion of susceptible individuals over the course of life using a mathematical model. Assuming that all unvaccinated individuals eventually contract varicella, our model considers that recruitment of new susceptible individuals occurs through births (*Finkenstädt & Grenfell, 2002*; *Wallinga, Teunis & Kretzschmar, 2003*) and that the size of the susceptible proportion of a given birth cohort subsequently declines either by vaccination or infection. The model imposes a few simplistic assumptions: (i) the first dose of varicella vaccine confers full protection by adulthood, (ii) following natural infection, acquired immunity persists until adulthood, and (iii) susceptible individuals experience time- and age-dependent annual risks of infection.

Reflecting these points, the balance equation of susceptible individuals is written as:

$$S_{t+1,1} = (1 - v_t)B_t - I_{t,1} \text{ for the first two years of life } (i = 1)$$
$$S_{t+1,i} = S_{t,i-1} - I_{t,i-1} \text{ for } i > 1, \tag{1}$$

where $t = 0, 1, 2, \dots$ and $i = 1, 2, 3, \dots$ denote the calendar year starting from 1990 and chronological age (in years), respectively. $S_{t,i}$ is the number of susceptible individuals in year $t$ and of age $i$, $v_t$ is the vaccination coverage in year $t$, $B_t$ is the number of newborns in year $t$, and $I_{t,i}$ represents the number of cases in year $t$ of age $i$. Using the number of susceptible individuals, the yearly reported number of new infections, $C_{t+1,i}$, is described by:

$$C_{t+1,i} = \delta_t I_{t+1,i} = \delta_t \gamma_{t,i} S_{t,i}, \tag{2}$$

where $\delta_t$ is the ascertainment factor in year $t$ and $\gamma_{t,i}$ is the annual risk of infection among susceptible individuals in year $t$ and of age $i$. Here, the ascertainment factor has been commonly referred to as "reporting rate" by time-series susceptible-infectious-recovered (TSIR) models (*Finkenstädt & Grenfell, 1998*). This is not precisely the reporting rate in that case numbers in surveillance report is not divided by actual number of cases (*Ciofi degli Atti et al., 2002*; *Marziano et al., 2018*), but the actual reporting rate in Japan is well known as difficult to be estimated explicitly, due to non-random distribution of medical and healthcare facilities that agreed to cooperate surveillance.

In the above model, vaccination coverage $v_t$ and the number of newborns $B_t$ are directly informed by empirical data, while $S_{t,i}$ and $I_{t,i}$ are reconstructed by estimating parameters $\delta_t$ and $\gamma_{t,i}$. First, the ascertainment factor $\delta_t$ was estimated using a linear regression analysis of the yearly number of newborns and cumulative number of reported cases in the corresponding birth cohort (see Supporting Material). Because the reporting system drastically changed in 1999 when the notification rules and corresponding laws were revised, ascertainment factor was estimated using a step function with parameters $\delta_A$ and $\delta_B$ representing the periods 1984–1999 and 2000–2015, respectively. Second, maximum likelihood estimation was conducted to estimate the annual risk of infection $\gamma_{t,i}$.

We assumed that the annual risk of infection changed every 5 years until 2009, then changed every 2 years after 2010 as the vaccination program was gradually expanded. To construct the likelihood function to estimate $\gamma_{t,i}$, age-specific annual incidence was assumed to follow a Poisson distribution.

$$L(\theta; r_{t,i}) = \prod_i \prod_t \frac{E(C_{t,i})^{r_{t,i}} \exp(-E(C_{t,i}))}{r_{t,i}!} \qquad (3)$$

where E(.) represents the expected number and $r_{t,i}$ is the reported (observed) number of cases in year $t$ and age group $i$.

Using parameter estimates, we reconstructed the number of susceptible individuals over time and by age. Subsequently, we computed the projected epidemiological dynamics of varicella until 2033. To account for parameter uncertainties, the 95% confidence intervals of projected notification rates were computed using the parametric bootstrap method.

## Data sharing statement

Annual varicella surveillance data by age group and demographic cohort of newborns are available as Online Supporting Material (Tables S1 and S2).

## Ethical considerations

The present study used publicly available information. Because no private information was used, ethical approval was not required.

## RESULTS

Figure 1A shows the age distribution of reported varicella cases from 1990 to 2019. The overall number of reported varicella cases declined by 76% from 2011 to 2019. Although incidence among preschool age children (age 1 to 4 years) continued to decrease after 2011, a similar decline was not observed among school-aged children. Varicella incidence among children aged 5 to 9 years decreased from 2011 to 2015, but it did not show a declining trend after 2015 (Fig. 1B). Interestingly, varicella incidence among teenagers aged 10 to 14 years increased every year after the initiation of a routine immunization program in 2014 (Fig. 1C). Varicella vaccine coverage gradually but steadily increased over time, with an abrupt increase observed after 2014. Although varicella vaccine coverage was only around 20% in the 1990s, coverage reached 90% by 2017 (Fig. 2).

Figure 3 shows the estimated annual risk of infection among susceptible individuals by age group and year. The annual risk of infection among pre-school children aged 1 to 4 years decreased starting in 2011. Declining annual risks of infection over time were subsequently observed among school-age children (age 5 to 9 years) after 2015. Conversely, the annual risk of infection among teenagers aged 10 to 14 years increased after 2011 (Fig. S2). The risk of infection was highest among 4-year-old children until 2013. After this time, the age of peak varicella risk increased, and in 2018/2019 the age group with the highest risk of infection became teenagers aged 10 to 14 years.

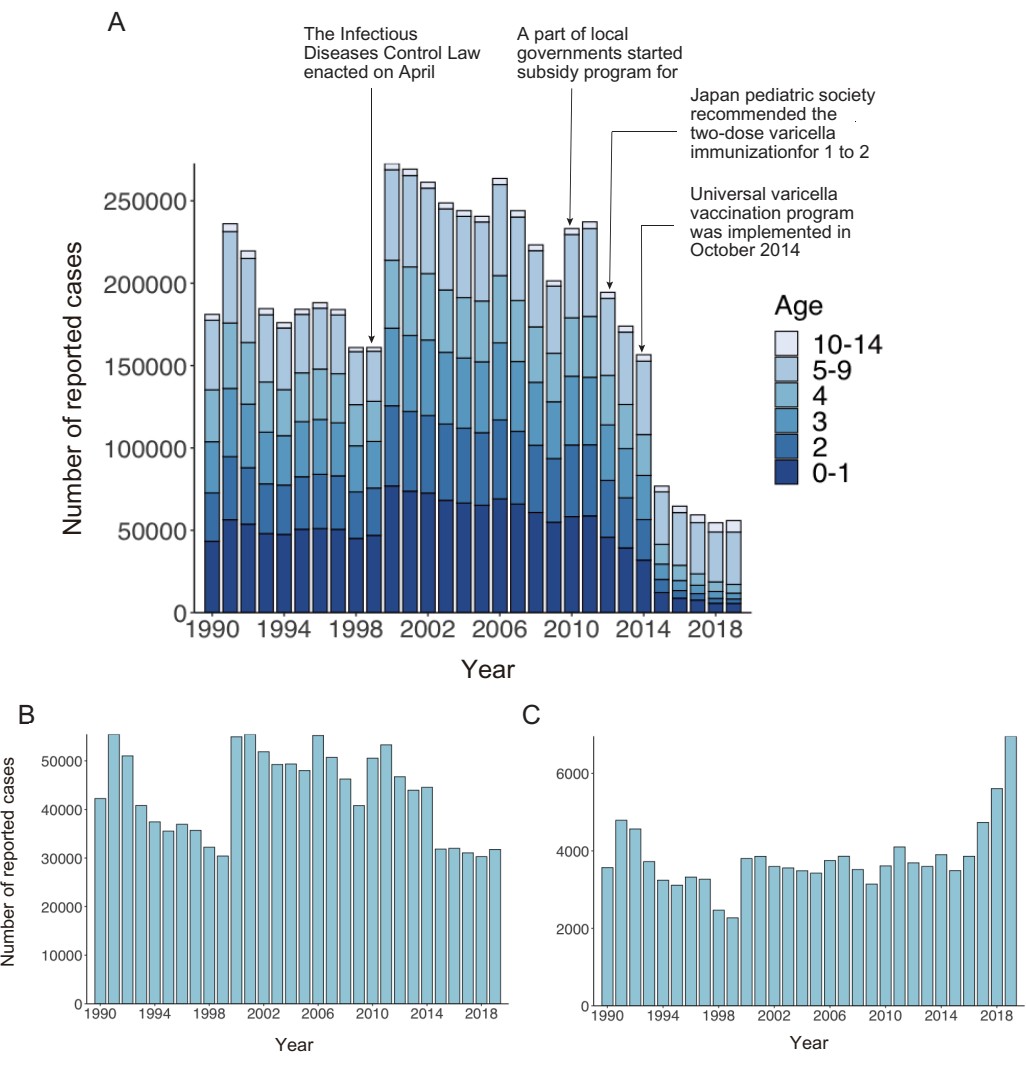

**Figure 1 Number of reported varicella cases in Japan, 1990–2019.** (A) Number of reported varicella cases by age. (B, C) Number of reported cases in individuals aged 5 to 9 years (B) and 10 to 14 years (C).

Comparisons of the observed and estimated numbers of reported varicella cases are shown in Fig. 4. While our model was kept very simple and tractable, it captured the empirically observed overall patterns of infection in all age groups, including those with both increasing and decreasing numbers of cases over time.

Figures 5A and 5B illustrate depletion of the susceptible population with advancing age, stratified by birth year. The susceptible population decreased over time, and the age-dependent decline in susceptibility was accelerated in pre-school age children following the introduction of routine immunization program. Figures 5C–5E show the long-term dynamics of the susceptible population by age over the past three decades and until 2033. Although the number of susceptible pre-school age children (age 1 to 4 years) decreased over time and remained at low level, numbers of susceptible children aged 5 years and older temporarily increased shortly after the initiation of routine immunization

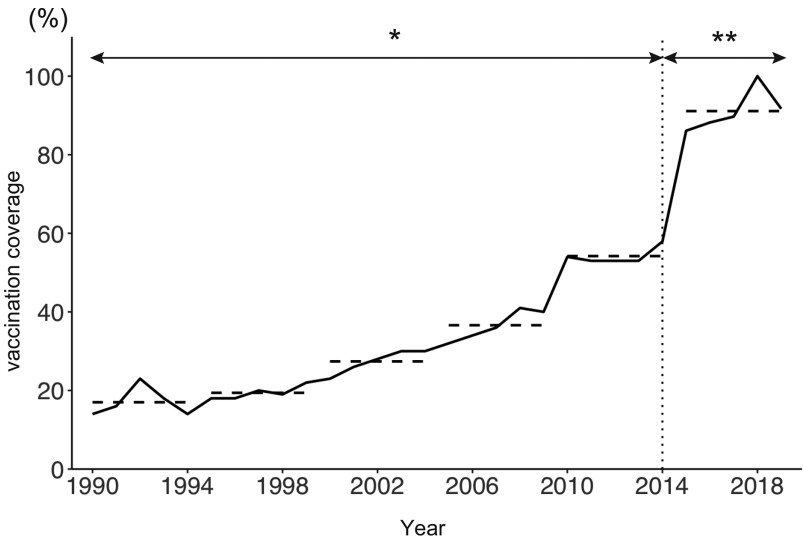

**Figure 2 Estimated varicella vaccine coverage in Japan, 1990–2019.** The solid line shows estimated varicella vaccine coverage and dotted horizontal lines show the 5-year average used for data analysis. The dotted vertical line indicates the year (2014) when universal varicella vaccine was implemented. *Estimated based on annual vaccine sales (vaccine doses sold/number of newborns in previous year). **Extracted from national epidemiological surveillance of vaccine-preventable diseases (NESVPD) database.

in 2014 (Fig. 5D). In older age groups, increased numbers of susceptible individuals were observed at later times (Figs. 5D and 5E). For example, the number of unvaccinated susceptible individuals showed a sharp peak among children aged 5 years in 2018. Aging of this unvaccinated birth cohort is seen every year, as illustrated by the damped, yet sharp, peak in 2023 for 10-year-old children.

## DISCUSSION

The present study evaluated the impact of the routine varicella immunization program in Japan on long-term varicella epidemiological dynamics using demographic and surveillance data as well as information on vaccination coverage. We first estimated the age- and time-dependent annual risks of infection among susceptible individuals. Using parameter estimates, we reconstructed and predicted varicella dynamics in Japan from 1990 to 2033. Our model successfully captured the age shift in susceptibility to varicella infection that occurred along with increased vaccine coverage; however, the most remarkable age shift in the susceptible population occurred *via* aging of the unvaccinated birth cohort. Following introduction of the routine immunization program, the risk of infection gradually increased among teenagers as unvaccinated susceptible individuals accrued in the corresponding age groups. The elevated age at infection has been observed in other countries including the USA (*Halloran et al., 1994*), and what is unique to Japan is that such change has been seen in a drastic manner following an introduction of universal vaccination program.

Two critical points should be taken from our study. Firstly, widespread vaccination had an immense impact on the epidemiological dynamics of varicella. The overall incidence of

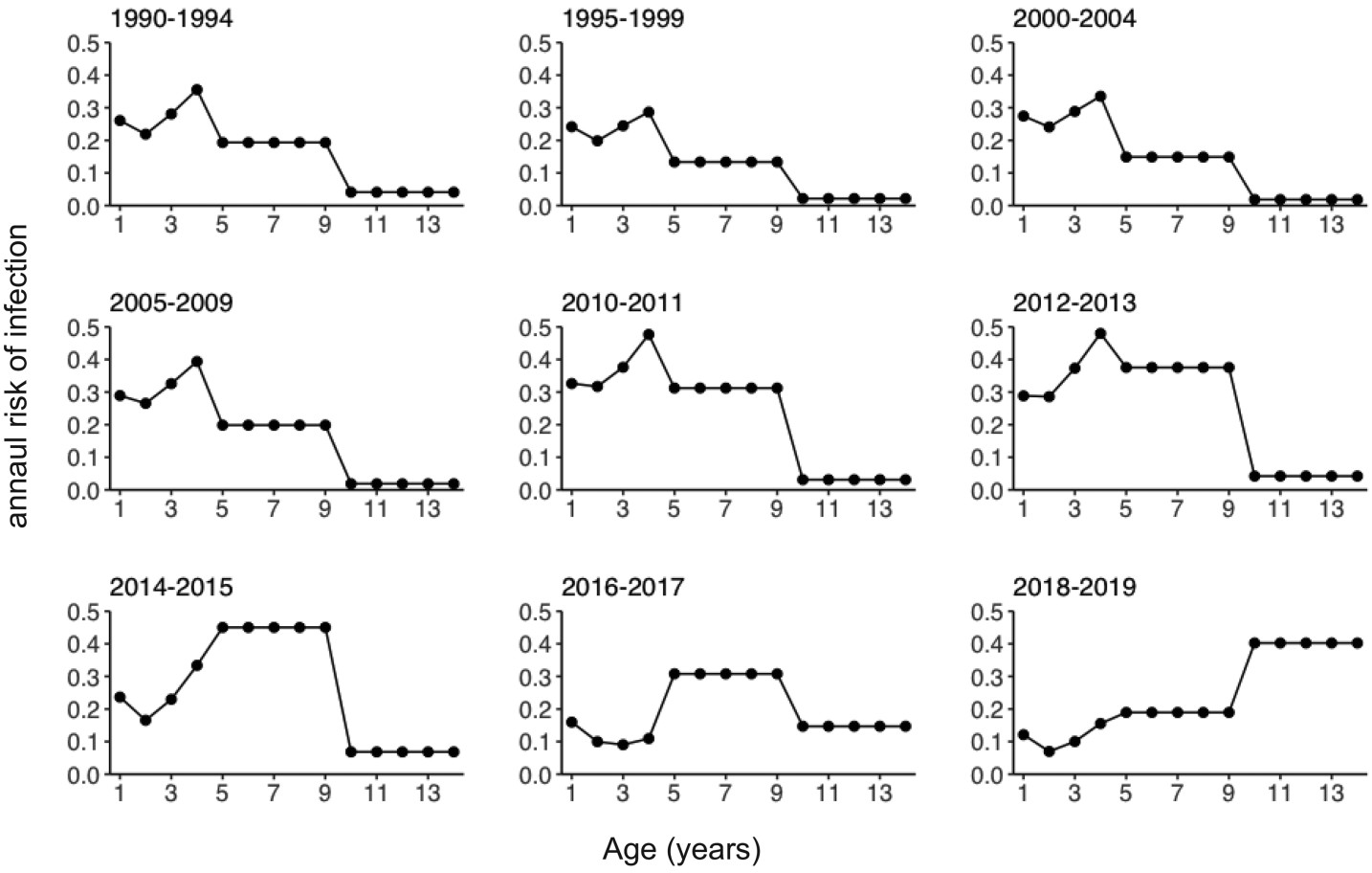

**Figure 3 Estimated annual risk of varicella infection among susceptible individuals.** Estimated annual risk of infection stratified by year.

varicella sharply declined, even before the initiation of routine vaccination when vaccination was voluntary and sporadically covered local government subsidies. The unvaccinated birth cohort who had never contracted varicella remained susceptible and aged without contracting varicella. Using modeling techniques, we successfully reconstructed the dynamics of the susceptible population over time. The number of susceptible individuals decreased in all age groups starting in the 1990s. However, there was a short-term increase in the number of susceptible children under 5 years of age shortly after the introduction of routine vaccination. Subsequently, a short-term increase in the number of susceptible individuals was observed in older age groups. Although susceptibility eventually decreased, many unvaccinated individuals born from 2010 to 2014 (*i.e.*, shortly before implementation of the routine immunization program) remained susceptible until their teenage years. Our model projected that this over-represented susceptible cohort may remain identifiable until around 2027.

Secondly, it should be noted that the annual risk of infection increased with age, especially among school-age children. The annual risk of infection among children aged 5 to 9 years elevated during the period of varicella vaccine expansion around 2010, which

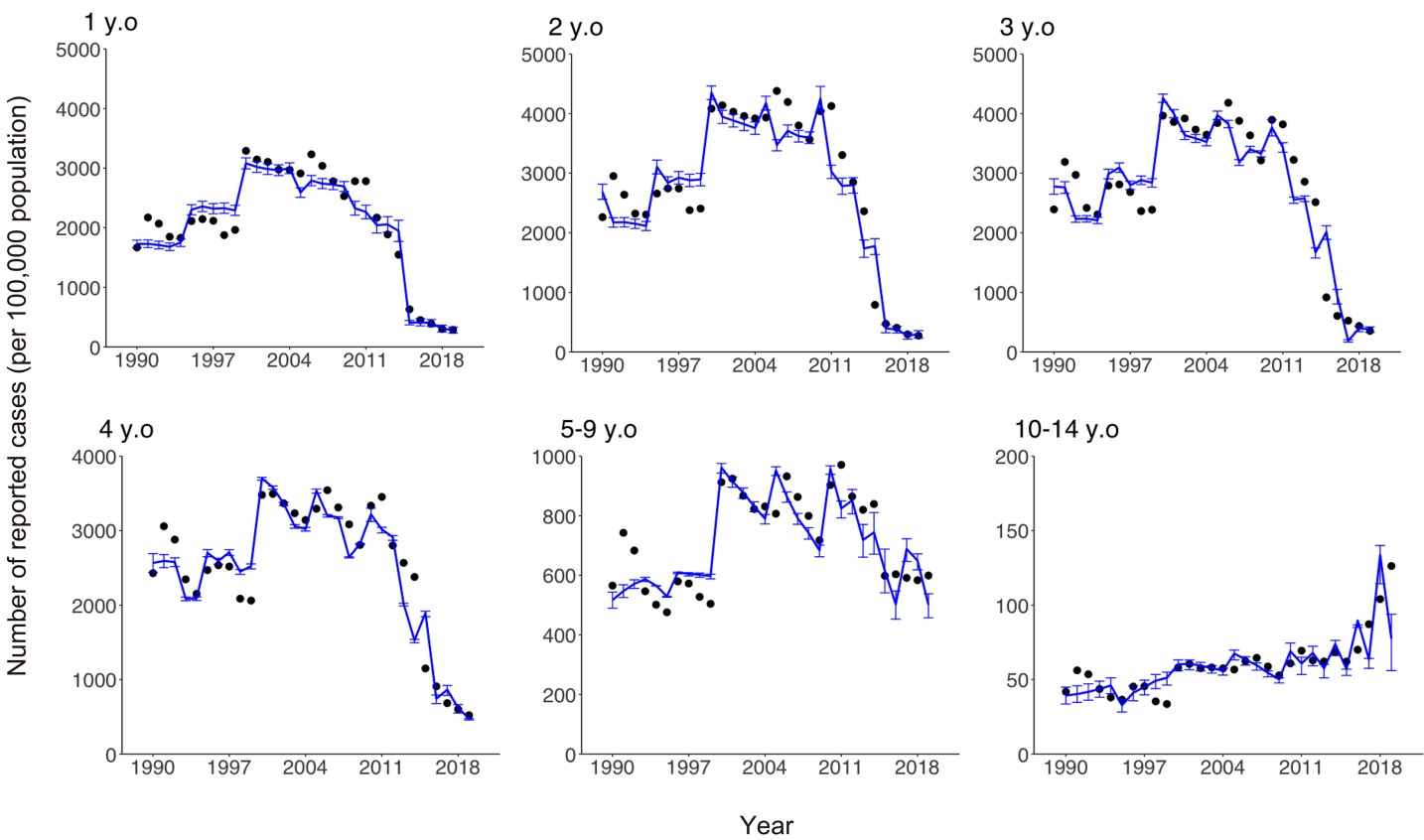

**Figure 4 Comparison of observed and estimated numbers of reported varicella cases by age.** Black dots show numbers of reported cases. Blue lines show estimated numbers of reported cases and blue error bars indicate 95% confidence intervals.

was not in line with declining birth rate and subsidized vaccination (at moderate coverage) at local municipality levels. We speculate that the number of susceptible population increased before 2014 among children aged 5–9 and this increase in susceptibility resulted in the increased risk of infection among them. A published study from France that explored pre-school children attributed similar increase to an increased social mixing among pre-school children (*Marziano et al., 2018*). The annual risk of infection among children aged 5 to 9 years subsequently declined following implementation of the routine vaccination program starting in 2014. Among teenagers, annual risk of infection has continued to increase since 2010. The annual risk of infection can increase either by increased transmissibility (or transmission rate) or increased prevalence of infection. Considering that the age of peak susceptibility temporarily increased during the period of varicella vaccine expansion, we speculate that an increase in the number of infectious individuals resulted in the elevated risk of infection among teenagers (*e.g.,* increased chance of transmission in school settings).

The accumulation of susceptible adolescents implies the potential for varicella outbreaks in middle or high schools in the near future in Japan. Modeling has also suggested the potential for post-honeymoon epidemics for other vaccine preventable diseases such as measles (*Metcalf et al., 2020*). In the case of varicella, school outbreaks among highly

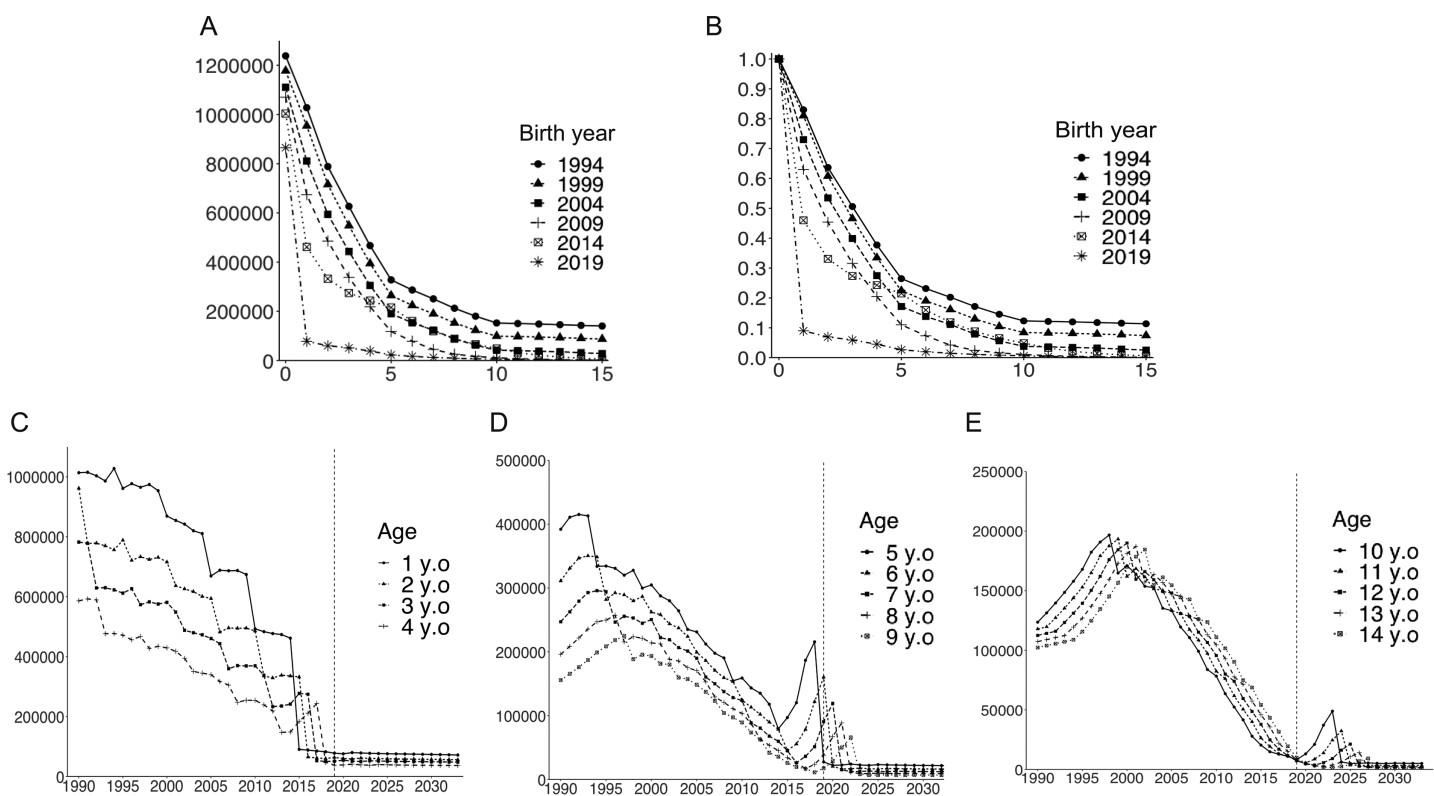

**Figure 5 Temporal dynamics of the varicella-susceptible population.** (A) Estimated number of susceptible individuals stratified by birth year. (B) Estimated proportion of susceptible individuals (estimated number of susceptible individuals/number of newborns in birth cohort) stratified by birth year. (C–E) Estimated number of susceptible individuals by age. Vertical dotted lines indicate the last year for the analyzed datasets (2019).

vaccinated populations have been reported in the United States, where two-dose routine vaccination was implemented in 2007 (*Gao et al., 2010*; *Lopez, Zhang & Marin, 2016*). Although most outbreaks were observed among children aged 5 to 9 years, several outbreaks occurred among high school students who are at risk for severe varicella infection (*Boëlle & Hanslik, 2002*). A study analyzing active surveillance data for varicella outbreaks from 2012 to 2015 concluded that most outbreak-associated infections occurred in under-vaccinated children who were unvaccinated or received only single-dose vaccination (*Lopez et al., 2019*). The same study found that outbreak sizes were smaller among students because of the high proportion of two-dose vaccine recipients.

Varicella vaccination has been shown to alleviate the severity of the disease. A systematic review of studies published from 1974 to 2016 concluded that severe breakthrough varicella is rare, but can occur (*Leung, Broder & Marin, 2017*). To avoid varicella outbreaks as well as severe infection, ensuring primary and/or booster vaccination of adolescents and young adults may need to be considered at some point in the future, as was the case for rubella to minimize risk of congenital rubella syndrome (*Kinoshita & Nishiura, 2016*). The observed phenomena was likely associated with the increased clinical severity of varicella among teenagers compared with younger children, while vaccination coverage among adolescent remained insufficient. Regular observation of

seroepidemiological data is also vital to identify birth cohorts at high risk of varicella infection. If the cost associated with adolescent cases is substantial, or if the country aims to eliminate varicella, the possible solution would include supplementary immunization program among adolescents. To accomplish objective judgement of such policy option, regular monitoring of susceptibility, especially among unvaccinated birth cohorts not subjected to two-dose universal vaccination, will be required for the next several decades.

Several limitations of this study must be noted. Firstly, we employed a birth cohort model, and the model structure may have been oversimplified. There are inevitable trade-offs between model complexity and potential to provide biological insights. Despite its simplistic nature, the model used in the present study successfully captured the aging of unvaccinated susceptible individuals and increased risks of infection among adolescents, allowing future projections of varicella dynamics. Secondly, we assumed that single-time varicella vaccination in the model successfully prevented infection. One randomized controlled study comparing single-dose and two-dose varicella vaccination reported that vaccine efficacy after 10-year follow-up was 94.4% for single-dose and 98.3% for two-dose vaccination (*Kuter et al., 2004*). Thirdly, we did not consider waning immunity over time since vaccination. Duration of protection to varicella is not well understood (*SAGE Working Group on Varicella & Herpes Zoster Vaccines, 2014*). However, a study conducted during the period of single-dose vaccination in the United States showed that the risk of moderate or severe varicella among children who had received varicella vaccines at least 5 years previously was 2.6 times higher than among children who had been vaccinated less than 5 years previously (*Chaves et al., 2007*).

The present study did not investigate the epidemiology of herpes zoster. An increased incidence of herpes zoster following initiation of the universal varicella vaccination program has been reported in Japan (*Toyama, Shiraki & Miyazaki Dermatologist Society, 2018*). *Toyama, Shiraki & Miyazaki Dermatologist Society (2018)* concluded that declining varicella incidence among children reduced exogenous boosting events among their parents. As a result, herpes zoster incidence increased among adults, consistent with the Hope-Simpson hypothesis (*Hope-Simpson, 1965*; *Thomas, Wheeler & Hall, 2002*). Another possible explanation for the increase in herpes zoster incidence in Japan is the consequence of the declining in birth rate and the ageing society. Modeling study conducted in Spain (*Marziano et al., 2015*) and Germany (*Horn et al., 2018*) indicated that the demographic changes reduced the opportunities exogenous boosting, resulted in the increase in herpes zoster incidence, especially in elderly population. Future studies on this subject could explore these possible mechanisms along with empirical data.

The present study did not account for the potential effects of public health and hygiene measures implemented in response to the coronavirus disease 2019 pandemic on varicella dynamics. A study using a time-series Susceptible-Infected-Recovered (TSIR) model indicated the potential for future large outbreaks of endemic diseases, such as respiratory syncytial virus (RSV) and seasonal influenza, because of increasing numbers of susceptible individuals during controlled periods (*Baker et al., 2020*). In line with this hypothesis, increased numbers of RSV infections were reported in the 2021 season in Tokyo, Japan

(*Ujiie et al., 2021*). Similar effects may be observed for other infectious diseases, including varicella. This possibility will need to be examined in a future study.

While significant future work remains, the present study successfully evaluated the impact of routine varicella immunization on transmission dynamics in Japan. Direct and indirect effects of vaccination drastically reduced varicella incidence, but the program resulted in subgroups of unvaccinated susceptible individuals and increased annual risks of infection among adolescents, indicating the potential for future outbreaks. Regular monitoring of susceptibility, especially among unvaccinated birth cohorts not subjected to two-dose universal vaccination, will be required for the next several decades.

## ACKNOWLEDGEMENTS

We thank Edanz Group for editing a draft of this manuscript.

### Funding

This study was supported by funding from Health and Labor Sciences Research Grants (19HB1001, 19HA1003, 20CA2024, 20HA2007, and 21HB1002 to H.N.), the Japan Agency for Medical Research and Development (JP20fk0108140 and JP21fk0108612 to H.N.), the Japan Society for the Promotion of Science KAKENHI (A.S: 19K24159, H.N: 17H04701 and 21H03198), the Inamori Foundation, the GAP Fund Program of Kyoto University, the Japan Science and Technology Agency CREST program (JPMJCR1413 to H.N.), and the SICORP program (JPMJSC20U3 and JPMJSC2105 to H.N.). The funders had no role in study design, data collection and analysis, decision to publish, or preparation of the manuscript.

### Grant Disclosures

The following grant information was disclosed by the authors:
Health and Labor Sciences Research: 19HB1001, 19HA1003, 20CA2024, 20HA2007, and 21HB1002 to H.N.
Japan Agency for Medical Research and Development: JP20fk0108140 and JP21fk0108612 to H.N.
Japan Society for the Promotion of Science KAKENHI: A.S: 19K24159, H.N: 17H04701 and 21H03198.
Inamori Foundation.
GAP Fund Program of Kyoto University.
Japan Science and Technology Agency CREST program: JPMJCR1413 to H.N.
SICORP program: JPMJSC20U3 and JPMJSC2105 to H.N.

### Competing Interests

Hiroshi Nishiura is an Academic Editor for PeerJ.

## Author Contributions

- Ayako Suzuki performed the experiments, analyzed the data, prepared figures and/or tables, authored or reviewed drafts of the paper, and approved the final draft.
- Hiroshi Nishiura conceived and designed the experiments, analyzed the data, authored or reviewed drafts of the paper, and approved the final draft.

## Data Availability

The raw data of sentinel-based surveillance record of chickenpox cases in Japan and recorded number of newborns are available in the Supplementary Tables.

## Supplemental Information

Supplemental information for this article can be found online at http://dx.doi.org/10.7717/peerj.12767#supplemental-information.

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
