# Peer review of "Reconstructing the transmission dynamics of varicella in Japan: an elevation of age at infection"

_PeerJ, doi:10.7717/peerj.12767_

## Round 0.1 · original submission · Major Revisions

To clarify the issues raised by the reviewers and also to improve the content of the manuscript, I would like to invite the authors to revise their manuscript.

·

Basic reporting

-

Experimental design

-

Validity of the findings

-

Additional comments

This is an epidemiological study to follow the impact of universal varicella vaccination program in Japan, initiated in 2014. Two doses of vaccine are given between ages 12 and 36 months. This information is key to the findings and should be given also in the Abstract. Also the level of coverage should be mentioned in the Abstract.

The title of the paper is complicated. The main observation is a shift in age from young children to older children and teenagers. This should be reflected in the title.

The Introduction is long. It could be shortened by deleting the last but one paragraph discussing herpes zoster. This study does not address the incidence of zoster. Moreover, for a full review of the subject the one paragraph is not enough. Likewise, the corresponding paragraph on zoster in Discussion should be deleted.

The data indicate that following varicella UMV in Japan the age of catching chicken pox has shifted from young children to teenagers. This is associated with a higher risk of complications, although this study does not provide any numbers in this regard. The age shift was observed in the US twenty years ago. The main reason was insufficient vaccine coverage, as it probably is in Japan.

Still, the present experience in Japan is important and worth reporting. It should be noted, however, that the increase in absolute numbers is relatively small. There are five Figures which is a lot. Of there, perhaps Fig. 3 could be deleted and shifted to supplementary material.

The remedy is obvious, but is not much discussed. The start of the program in 2014 could and should have included a catch-up vaccination up to the age of 12 years, as has been done in some countries. At the present stage, a remedy could a separate teenager vaccination program which many countries practice. While the authors’ model predicts an increase of varicella in teenagers for years to come, this does not need to happen. The authors should be bolder in their discussion about this issue.

Reviewer 2 ·

Basic reporting

The paper is well written and clear in most of its parts.

Figures presented are relevant, well labelled and described and raw data are supplied.

In the Introduction the authors provide a satisfactory background on the main epidemiological features of varicella zoster virus and on the public health issue related to varicella vaccination and its possible negative impact on herpes zoster incidence.

Experimental design

The research question is well defined, however it is not clear to me if the authors fill some gap of knowledge on varicella dynamics in the presence of vaccination. Indeed, the epidemiological dynamics of varicella observed in Japan and described by the authors are in line with what observed in other geographical settings (Taiwan, United States, etc).

Methods description can be improved by addressing my suggestions/concerns in the section "Additional comments".

Validity of the findings

No comment

Additional comments

- I have one major methodological concern. I am a little bit confused by the definition of reporting rate used in this study. In my experience, the reporting rate is commonly defined as the proportion of infections occurring within a given population, that are detected by the surveillance system. In the case of this work, I would suggest that the numerator should be the number of varicella cases reported to the NESID surveillance system and the denominator should be the real number of varicella infections (either detected or undetected) occurring in the same period within the population “served” by the pediatric sentinel sites included in the NESID system. The value of the reporting rate can be estimated through mathematical models or other approaches (see e.g. doi:10.1017/S0950268802006878 ; or https://doi.org/10.1371/journal.pcbi.1006334).
If data needed to estimate the reporting rate in Japan are not available, one could make the simplifying assumption that all varicella infections occurring in the population served by the sentinel sites are detected by the system, i.e. the reporting rate is 100%.
Under this assumption, the total number of cases reported to the surveillance system could be “projected” to the whole Japanese population as:
casesJAPAN=casesNESID*(populationJAPAN/populationNESID)
where (populationNESID/populationJAPAN)=10% and the whole analysis could be performed directly at the national level.
I find it difficult to interpret why the authors compute the reporting rate as the ratio between cases reported within a given period and the number of newborns (I think at the national level) during the same period. As far as I understand, the underlying assumption is that all individuals will develop varicella at some point. My concern is that this “certain point” could also lie outside the time interval considered. In particular, I think that changing demography could play a role, especially in the presence of varicella vaccination that can lead to a shift of infections towards older ages and in a period characterised by changes in the birth rate.

- Fig 3A: In the discussion you comment on the increased risk at infection among children aged 5-9yrs before 2011. Do you have an explanation for the increased risk among children of younger ages (<5 years)? In this period the birth rate in Japan was declining and vaccination was in place (even though the coverage was relatively low). This should result in a shift of infections towards older age groups (e.g. 5-9yrs). I would thus expect a declining risk among young children. A previous study (https://doi.org/10.1371/journal.pcbi.1006334) explained the increase of varicella incidence observed among French pre-school children in a period characterised by a stable birth rate and in the absence of vaccination, with an increase of social-mixing in these age groups (possibly ascribable to increased attendance rate to nurseries and pre-schools). Could this represent a possible explanation also for Japan?

Minor points:
- Abstract, line 30: “The unvaccinated birth cohort who had never contracted varicella remained susceptible over this period”. Do you refer to the result expressed in Discussion as: “A substantial number of unvaccinated individuals born before the routine immunization era remained susceptible and aged without contracting varicella.”? If yes, please consider rephrasing the sentence of the abstract, because the message is not clear.
- l. 88: when the cumulative number susceptible individuals  missing “of”
- Introduction (l.109): I think that 2024 should be changed to 2033, right?
- L.149. In equation 1 the variable C represents varicella infections (all infections, detected or not) while in Equation 2, the variable I represents infections reported to the surveillance system. Conventionally, “infections” denote all infections (including those not reported to the surveillance system), while “cases” is more widely used to denote reported cases. I would suggest switching the notation, i.e. to use in Equation 1 the notation I and refer to this variable as infections, and to use notation C in Equation 2 and to refer to this variable as “cases” or “reported cases”.
- L.164: please report the likelihood function used to estimate the annual risk of infection.
- Fig 1 and 4 would be probably more informative when plotting incidence rates, instead of absolute numbers (if available).
- L. 243-245: the second part of this sentence sounds like a validation of the model, but if I understand well, the annual risk of infection is estimated using age-specific incidence rate. If this is the case, please remove.
- Discussion: previous studies (e.g. https://doi.org/10.1098/rspb.2014.2509 or https://doi.org/10.1186/s12916-017-0983-5 ) suggest that an increase of herpes zoster incidence could be expected also as a consequence of demographic changes, and in particular as a consequence of declining historical trends in the birth rate. You could consider adding a sentence in the Discussion (e.g. after l.290-293) stating that part of the observed increase could also be related to declining birth rate.

---

## Round 0.2 · accepted · Accept

Both reviewers are satisfied with the authors response. However, reviewer #2 has a very minor comment as follows: "I suggest the authors to replace the term "elevation" in the text and in the title of the manuscript with one of the following "increase/growth/rise". The authors could make these changes during proof correction stage. I congratulate the authors for their work.

·

Basic reporting

-

Experimental design

-

Validity of the findings

-

Additional comments

-

Reviewer 2 ·

Basic reporting

No comment

Experimental design

No comment

Validity of the findings

No comment

Additional comments

I am satisfied with the authors response.

I have only one minor comment: I suggest the authors to replace the term "elevation" in the text and in the title of the manuscript with one of the following "increase/growth/rise".